# Aldh2 Attenuates Stem Cell Factor/Kit-Dependent Signaling and Activation in Mast Cells

**DOI:** 10.3390/ijms20246216

**Published:** 2019-12-10

**Authors:** Do-Kyun Kim, Young-Eun Cho, Byoung-Joon Song, Toshihiro Kawamoto, Dean D. Metcalfe, Ana Olivera

**Affiliations:** 1Mast Cell Biology Section, Laboratory of Allergic Diseases, National Institute of Allergy and Infectious Diseases (NIAID), National Institutes of Health (NIH), Bethesda, MD 20892, USA; dmetcalfe@niaid.nih.gov; 2Section of Molecular Pharmacology and Toxicology, Laboratory of Membrane Biochemistry and Biophysics, National Institute on Alcohol Abuse and Alcoholism (NIAAA), NIH, Bethesda, MD 20892, USA; yecho@andong.ac.kr (Y.-E.C.); bj.song@nih.gov (B.-J.S.); 3Occupational Health Research and Development Center, Japan Industrial Safety and Health Association, Tokyo 108-0014, Japan; t-kawamoto@jisha.or.jp

**Keywords:** Aldh2 deficiency, mast cells, Kit, Shp-1, mast cell activation, proliferation

## Abstract

Mitochondrial aldehyde dehydrogenase (ALDH2) metabolizes endogenous and exogenous aldehydes and protects cells against oxidative injury. Inactivating genetic polymorphisms in humans are common and associate with alcohol flush reactions. However, whether mast cell Aldh2 activity impacts normal mast cell responses is unknown. Using bone marrow-derived mast cells from *Aldh2* knockout mice, we found evidence for a role of mast cell Aldh2 in Kit-mediated responses. Aldh2-deficient mast cells showed enhanced Kit tyrosine kinase phosphorylation and activity after stimulation with its ligand (stem cell factor) and augmentation of downstream signaling pathways, including Stat4, MAPKs, and Akt. The activity of the phosphatase Shp-1, which attenuates Kit activity, was reduced in *Aldh2^−/−^* mast cells, along with an increase in reactive oxygen species, known to regulate Shp-1. Reduced Shp-1 activity concomitant with sustained Kit signaling resulted in greater proliferation following Kit engagement, and increased mediator and cytokine release when *Aldh2^−/−^* mast cells were co-stimulated via Kit and FcεRI. However, FcεRI-mediated signaling and responses were unaffected. Therefore, our findings reveal a functional role for mast cell intrinsic Aldh2 in the control of Kit activation and Kit-mediated responses, which may lead to a better understanding of mast cell reactivity in conditions related to ALDH2 polymorphisms.

## 1. Introduction

Aldehyde dehydrogenase 2 (Aldh2) is a mitochondrial enzyme that protects cells from biogenic aldehydes accumulated through metabolism, and the most efficient isoenzyme within the family of ALDH enzymes to remove toxic acetaldehyde derived from the metabolism of alcohol [1]. Some aldehyde intermediates are highly reactive and modify proteins, cause protein aggregation, and produce reactive oxygen species (ROS), and Aldh2 thus plays a protective role in cells during oxidative stress [1,2,3]. A genetic polymorphism (rs671) in *ALDH2 (ALDH2*2)* is the most common single point mutation in humans, present in approximately 40% of Eastern Asian populations [1,4]. This polymorphism causes a severe reduction in ALDH2 activity, even in heterozygous individuals, through a dominant negative effect and is associated with conditions such as alcohol flush syndrome [5], manifested by facial flushing, headaches, nausea, dizziness, and cardiac palpitations after the consumption of alcoholic beverages [1]. Flushing has been linked to the activation of mast cells [6,7] and in alcohol flushing mast cell involvement is suggested by reports showing that the metabolite of alcohol acetaldehyde causes mast cell degranulation and increases histamine release [8,9,10], and by the improvement of alcohol flushing by antihistamine treatment [11].

Mast cells are characterized by the expression of FcεRI, the high-affinity IgE receptor [12], and their activation via this receptor by multivalent antigen (Ag) results in the release of granule-associated mediators and *de novo* synthetized cytokines [12,13]. FcεRI stimulation in tissues occurs in the context of signals derived from Kit, the receptor for the stem cell factor (SCF) which is produced in tissues and enhances mast cell responses to IgE/Ag and other mast cell stimulants. In addition, Kit is critical for mast cell proliferation and survival [14,15]. Therefore, understanding the factors that impact Kit signaling in mast cells is important for understanding mast cell responsiveness.

The activation of mast cells causes transient increases in ROS that regulate mast cell signaling and responses [16,17,18,19]. Given the reported role of mitochondrial Aldh2 in the regulation of oxidative stress [1,3], and the associations between Aldh2, mast cells, and alcohol-induced pathologies, we sought to investigate whether Aldh2 activity plays a role in regulating mast cell behavior following FcεRI and Kit activation.

In this report, we present evidence that bone marrow-derived mast cells (BMMCs) from mice with a genetic deletion in *Aldh2* have increased proliferation and IL-6 production after stimulation with SCF, and when co-stimulated with SCF and IgE/Ag, show enhanced mediator release. Kit phosphorylation and the activation of downstream signaling molecules that are critical for mast cell responses [15,20] were also enhanced in Aldh2-deficient BMMCs after SCF stimulation. These effects were associated with an increase in ROS levels and a reduction of activity of the Src homology domain 2-containing protein tyrosine phosphatase 1 (Shp-1), which is a negative regulator of signaling by Kit. Our findings are consistent with the conclusion that Aldh2 plays a role in the negative regulation of Kit signaling and may provide insight into the regulation of mast cell responsiveness in relation to alcohol-associated flushing.

## 2. Results

### 2.1. Aldh2 Deficiency Enhances Mast Cell Proliferation

After 4 weeks in culture, >97% of both *Aldh2*^+/+^ and *Aldh2*^−/−^ bone marrow cells (*n* = 5 independent cultures/genotype) were positive for Kit and FcԑRI, characteristically expressed in mast cells. The levels of expression of Kit and FcԑRI, as determined by FACS analyses, were similar in mast cells from either genotype (Figure 1A). However, the number of total cells in the cultures derived from *Aldh2*^−/−^ mice was ~2-fold higher than those derived from *Aldh2*^+/+^ mouse marrows after 4 to 5 weeks in culture (Figure 1B). Further, when 5-week-old mature BMMCs were plated at the same density, *Aldh2^−/−^* cells continued to increase in number at a higher rate than *Aldh2^+/+^* BMMCs (Figure 1C). To further document that the proliferation of *Aldh2^−/−^* mast cells was enhanced, we determined [^3^H]-thymidine incorporation in *Aldh2^+/+^* and *Aldh2^−/−^* BMMCs in response to SCF, a known growth factor for mast cells. [^3^H]-Thymidine incorporation in the presence of either 10 or 100 ng/mL SCF was significantly increased in *Aldh2^−/−^* compared with *Aldh2^+/+^* BMMCs (Figure 1D). Taken together, these results demonstrate that Aldh2 negatively regulates mast cell proliferation.

### 2.2. Responses to SCF Alone or in Combination with IgE/Ag Are Enhanced in Aldh2-Deficient BMMC, While Responses to FcεRI Stimulation Are Unaffected

We then determined whether Aldh2 may impact the FcεRI-dependent release of bioactive mediators from mast cells. The FcεRI-mediated degranulation (Figure 2A) and release of TNF-α and IL-6 (Figure 2B,C) were similar in BMMCs differentiated from *Aldh2^+/+^* and *Aldh2^−/−^* mice. Similarly, degranulation induced pharmacologically by thapsigargin, a drug that causes cytosolic Ca^2+^ increases by inhibiting Ca^2+^ uptake into intracellular stores (Figure 2D), was unaffected by Aldh2 deficiency.

We next examined the potential effect of Aldh2 deficiency on FcεRI-mediated responses in the presence of SCF, since a combination of FcԑRI and Kit-mediated signals is required for optimal responses of mast cells to IgE/Ag. Both degranulation (Figure 2E) and cytokine release (Figure 2F,G) induced by the co-stimulation of Kit and FcεRI were significantly enhanced in *Aldh2^−/−^* compared with *Aldh2^+/+^* BMMCs (compare also with Figure 2A–C). Furthermore, SCF in the absence of IgE-receptor stimulation also markedly enhanced the production of IL-6, a cytokine known to promote mast cell proliferation and responses [21], in *Aldh2^−/−^* compared to *Aldh2^+/+^* BMMCs (Figure 2H). These results suggest that Aldh2, by regulating SCF/Kit-mediated signaling events, may impact the optimal physiological responses of mast cells to IgE/Ag, even though it does not directly alter the responses to FcεRI.

### 2.3. Kit-Induced Phosphorylation Events Are Upregulated in Aldh2-Deficient BMMCs

Consistent with the lack of effect of Aldh2 on FcεRI-mediated degranulation and cytokine production, early signal events mediated by this receptor [12,15], such as Syk phosphorylation (Appendix A) and activation (Appendix A), as well as its downstream signals (i.e., linker for the activation of T cells (LAT) in Appendix A; and PLCγ1, Akt, Jnk, and Erk in Appendix A), were not altered in *Aldh2^−/−^* compared to *Aldh2^+/+^* BMMCs.

In contrast, the phosphorylation of key signaling pathways downstream of the Kit receptor, such as Stat4, Akt (Figure 3A), Erk, and Jnk (Figure 3B), was increased in *Aldh2^−/−^* BMMCs. As these signals mediate mast cell proliferation, degranulation, and cytokine release [20,22,23], and IL-6 promotes mast cell proliferation and function [21], the results provide an explanation for the observed enhanced proliferation in response to SCF and increased release of mediators in combination with IgE/antigen.

### 2.4. Deficiency in Aldh2 Results in Increased Kit Activation Concomitant with Reduced Shp-1 Activity and Increased ROS Levels

Since signaling downstream of Kit was enhanced in Aldh2-deficient mast cells, we surmised that Aldh2 may affect an early step in the activation of the receptor. SCF-induced phosphorylation of Kit in tyrosine 823, which occurs by the transphosphorylation of dimerized Kit by binding to SCF, was more pronounced and longer lasting in *Aldh2^−/−^* compared with *Aldh2^+/+^* BMMCs (Figure 4A). This occurred concomitantly with increases in Kit kinase activity assayed in Kit immunoprecipitates of SCF-activated BMMCs (Figure 4B).

The tyrosine phosphatase Shp-1 is an important regulator of Kit dephosphorylation and the termination of signaling. Therefore, we investigated the possibility that Aldh2 regulates Kit signaling via Shp-1. We immunoprecipitated Shp-1 in *Aldh2^+/+^* and *Aldh2^−/−^* BMMCs activated with SCF, and measured phosphatase activity in the immunoprecipitated immunocomplexes. Shp-1 activity in SCF-activated *Aldh2^−/−^* cells was significantly reduced compared to that in *Aldh2^+/+^* mast cells (Figure 4C). Since Aldh2 has been reported to regulate ROS in certain systems [1,2,3] and Shp-1 is sensitive to ROS [16], we also measured the increases in ROS induced by SCF in *Aldh2^+/+^* and *Aldh2^−/−^* BMMCs. We found that Aldh2 deficiency resulted in an increased production of ROS by SCF (Figure 4D), and that prevention of ROS actions by pretreatment with the general antioxidant TEMPO reversed the diminished activation of Shp-1 induced by SCF (Figure 4E). The data are thus consistent with the conclusion that Aldh2, through the regulation of ROS levels, impacts the activity of Shp-1.

## 3. Discussion

Mast cells are thought to play a role in alcohol flushing in individuals with a deficiency in the *ALDH2* gene, a critical enzyme in the metabolism of acetaldehyde and lipid aldehydes [1,5,10]. Mast cells can also cause flushing, as reported in mast cell activation disorders, where an abnormal activation of mast cells may occur [7]. As Aldh2 protects cells against oxidative stress which occurs transiently during the activation of mast cells, we explored a potential role for this enzyme in the regulation of mast cell activation and in the context of Kit activation. Herein, we implicate Aldh2 for the first time in the regulation of Kit signaling via Shp-1, and thus in enhanced mast cell activation when SCF is a co-stimulus.

Aldh2 deficiency in BMMCs did not alter FcεRI-induced signaling, such as Syk activation, Syk-dependent phosphorylation of LAT, and downstream events (Appendix A) required for the allergic activation of mast cells [23,24], and also did not alter the FcεRI-mediated release of bioactive mediators (Figure 2A–C). In contrast, lack of Aldh2 activity resulted in significantly enhanced mast cell degranulation and cytokine production after the co-stimulation of FcεRI with Kit. Since allergic stimulation of mast cells in tissues occurs presumably in the context of SCF-induced signals, our results suggest a predisposition in Aldh2-deficient mast cells to increased responsiveness. The enhanced responses in the presence of SCF stimulation were accompanied by greater and longer lasting Kit phosphorylation and tyrosine kinase activity, which resulted in the elevated phosphorylation of signaling proteins such as Erk and Jnk. The activation of these cascades may explain the augmented secretion of cytokines in response to SCF, as cytokine production in mast cells is typically regulated by activated MAPKs [15,25].

The increased Kit tyrosine kinase activity and enhanced activation of Akt, Erk, Jnk, and Stat4 signaling pathways in Aldh2-deficient BMMCs were associated with markedly reduced Shp-1 activity in these cells after SCF stimulation. The tyrosine phosphatase Shp-1 is known to bind Kit and negatively regulate Kit activity and Kit-mediated responses [26] and a reduction in Shp-1 due to degradation was proposed as a mechanism contributing to the oncogenic potential of mutated c-Kit [27]. Shp-1 deficiency, similar to our findings in Aldh2-deficient BMMCs, was also found to enhance mast cell mediator release, particularly after c-Kit and FcεRI-co-stimulation [28]. The reduced Shp-1 activity in Aldh2-deficient mast cells after SCF stimulation was concomitant with increases in intracellular ROS levels, which have been shown to cause the inhibition of Shp-1 [16], and was reversed by treatment with the general antioxidant TEMPO, suggesting a contribution of Aldh2 to the regulation of SCF-induced oxidative species with an impact on Shp-1 activity and thus Kit signaling. This contrasted with the lack of a role for Aldh2 and Shp-1 in FcεRI-mediated responses, since known targets for Shp-1 (LAT) and downstream effectors (PLCγ and MAPKs) [16,29] were not affected in *Aldh2^−/−^* mast cells after FcεRI-ligation (Appendix A). The reasons for this apparent specificity for SCF-induced Shp-1 activation require further investigation but may include differences in the species of ROS produced by Kit activation compared to FcεRI ligation, or differences in the sensitivity of Shp-1 pools associated with distinct signalosomes under different stimuli.

In addition to the findings described herein of the involvement of Aldh2 in Kit-mediated signaling and responses, other reports have indicated that this enzyme can be activated in mouse mast cells and the human mast cell line HMC-1 by the stimulation of Gα_0_/α_i_-linked G-protein-coupled receptors (GPCRs), such as adenosine receptors A2b/A3, histamine receptor 4, or sphingosine-1-phosphate type 1 receptor [30,31,32]. Enhanced Aldh2 activity by these receptors reduces the release of mast cell renin induced by ischemic conditions, also suggesting contributions of Aldh2 to mast cell responses in the context of heightened toxic aldehydes [8].

In summary, the data presented herein involves Aldh2 as a previously unrecognized regulator of mast cell proliferation and SCF/Kit-mediated mast cells responses via Shp-1 regulation. This may be relevant for mast cell activation and alcohol flushing responses observed in populations with *ALDH2*2* polymorphisms [6,7] and aid in the selection of management approaches.

## 4. Materials and Methods

### 4.1. Reagents

Antibodies and reagents were purchased from the following sources: antibodies against ALDH2 (sc-48838) and Kit (sc-13508) were from Santa Cruz Biotechnology (Dallas, TX, USA) and used at a dilution of 1:1000 for Western blotting. Anti-phospho-Syk (Tyr525/526) (#2711), -phospho-LAT (Tyr191) (#3584), -phospho-PLCγ1 (Y783) (#2821), -phospho-Stat4 (Tyr693) (#5267), -phospho-Akt (Thr308) (#9275), -phospho-SAPK/Jnk (Thr183/Tyr185) (#9251), and -phospho-p44/42 MAPK (Erk1/2) (Thr202/Tyr204) (#9101) antibodies were from Cell Signaling Technology (Beverly, MA, USA) and diluted to 1:2000 for Western blotting. Anti-β-actin (#A5316), used as a loading control at 1:5000 dilution, was purchased from Sigma-Aldrich (St Louis, MO, USA). Antibodies against Syk (sc-929), Shp-1 (sc-287), and Kit (sc-48838) for immunoprecipitations were from Santa Cruz Biotechnology. Cell culture reagents were from GIBCO/Invitrogen (Carlsbad, CA, USA). Dinitrophenyl (DNP)-specific monoclonal IgE and DNP-human serum albumin (BSA) were from Sigma (St. Louis, MO, USA).

### 4.2. Mice and BMMC Cultures

*Aldh2^−/−^* and *Aldh2^+/+^* C57BL/6 mice were generated and then kindly provided by Dr. Toshihiro Kawamoto, as previously described [33]. All mice were maintained and used in accordance with NIH guidelines and animal study proposal number LMBB-BS-1, which was approved by the NIAAA Animal Care and Use Committee (Approved March 1, 2017). BMMCs were isolated from the mouse bone marrows of 6 to 8-week-old mice and cultured for 4–8 weeks in RPMI 1640 containing 2 mM L-glutamine, 0.1 mM nonessential amino acids, 100 U/mL penicillin, 100 µg/mL streptomycin, 1 mM sodium pyruvate, 1 mM HEPES, and 10% FBS and 10 ng/mL IL-3 [34]. The purity of mast cells in the cultures was monitored by assessing the percentage of cells expressing Kit and FcεRI by flow cytometry. Functional studies were conducted on cultures containing >95% double-positive mast cells. Flow cytometry analysis of Kit and FcεRI expression was performed as previously described [35] in an LSR II flow cytometer (BD Biosciences, Franklin Lakes, NJ, USA) and evaluated using FlowJo software (Tree Star Inc., Ashland, OR, USA).

### 4.3. Cell Proliferation Assays

BMMC proliferation assays were performed as previously described [36]. Briefly, cells were resuspended in DMEM + 10% FBS media without cytokines for 6 h and plated in triplicate in 96-well plates (3 × 10^4^ cells/well) in the presence of the indicated concentrations of mouse recombinant SCF for 24 h. [^3^H]-Thymidine (1 μCi) was added to each well for 6 h. Cells were collected using a Cell Harvester (Skatron, Sterling, VA, USA), and acid-insoluble [^3^H]thymidine in cells after precipitation with 5% trichloroacetic acid was determined using a Beta Plate Liquid Scintillation Counter (PerkinElmer Life Sciences, Boston, MA, USA).

### 4.4. Measurement of Degranulation and TNF-α and IL-6 Levels

For experiments where mast cells were stimulated, BMMCs (10^7^ cells/well) were incubated overnight in cytokine-free media containing mouse monoclonal anti-dinitrophenyl (DNP)-IgE (Sigma-Aldrich) (100 ng/mL). On the following day, cells were rinsed with HEPES-BSA buffer (10 mM HEPES [pH 7.4], 137 mM NaCl, 2.7 mM KCl, 0.4 mM Na_2_HPO_4_·7H_2_O, 5.6 mM glucose, 1.8 mM CaCl_2_·2H_2_O, 1.3 mM MgSO_4_·7H_2_O, and 0.04% BSA) and then stimulated with 25 ng/mL DNP-HSA (Ag), in the presence or absence of 100 ng/mL SCF. For degranulation, cells were stimulated with antigen in HEPES-BSA buffer for 30 min, and IgE-primed BMMCs for cytokine production were stimulated with Ag for 8 h in complete media without FBS. Degranulation was determined based on a measurement of the release of the granule marker β-hexosaminidase, as previously described [37], and calculated as percentages of total β-hexosaminidase content found in the supernatants after challenge. IL-6 and TNFα levels in the supernatants were determined by ELISA kits from R&D Systems (Minneapolis, MN, USA).

### 4.5. Immunoprecipitation and Western Blotting Analysis

IgE-primed BMMCs were stimulated with Ag in HEPES-BSA buffer for 0, 3, 7, and 15 min; washed twice with PBS; and lysed for 10 min on ice with lysis buffer containing 10 mM Tris-HCl (pH 8.0), 150 mM NaCl, 1 mM EDTA, 1 mM Na_3_VO_4_, 0.5 mM PMSF, 5 mg/mL aprotinin, 5 mg/mL leupeptin, complete protease inhibitor cocktail (Roche, Indianapolis, IN, USA), and 1% NP-40. Samples were then clarified at 13,000 *g* for 5 min at 4 °C. Protein content in the lysates was determined using a Bradford assay (Bio-Rad, Hercules, CA, USA) and equal amounts of protein (1 mg in mL lysis buffer) were used for immunoprecipitation of the indicated tyrosine kinases. Cell lysates were incubated at 4 °C with the indicated specific antibodies with gentle rocking for 4 h and the immunocomplexes were captured with protein A/G–agarose overnight at 4 °C. The agarose beads were washed five times with washing buffer (lysis buffer diluted 1:10 in PBS (pH 7.4)), resuspended in 2X Laemmli buffer, and then boiled for 10 min.

For Western blotting analysis, equal amounts of proteins (20 µg) were separated by SDS-PAGE using 4–12% gels and then transferred onto nitrocellulose membranes. The membranes were blocked with 5% BSA in Tris-buffered saline for 1 h, and then incubated with specific primary antibodies overnight at 4 °C. Immunoreactive bands were detected using infrared dye-conjugated secondary antibodies (LI-COR Biosciences; Lincoln, NE, USA) (1:20,000), and were imaged and quantified using an Odyssey imager (LI-COR Biosciences).

### 4.6. In Vitro Kinase Assay

Kit and Syk were immunoprecipitated from 1 mg whole-cell lysates of BMMCs stimulated or not stimulated with Ag (25 ng/mL) or SCF (100 ng/mL) in HEPES-BSA buffer for 0, 3, 7, and 15 min. Immunoprecipitates were assayed for kinase activity by using the ELISA-based Universal Tyrosine Kinase Assay Kit (Gen Way, San Diego, CA, USA), according to the manufacturer’s instructions. One unit of tyrosine kinase represents the incorporation of 1 pmol of phosphate into the substrate (a p34cdc2 peptide fragment of about 6–20 amino acids) per minute.

### 4.7. Measurement of Shp-1 Phosphatase Activity

Shp-1 was immunoprecipitated from BMMCs stimulated with 100 ng/mL SCF in HEPES-BSA buffer for 0, 3, 7, and 15 min. Shp-1 activity in the immunoprecipitates was measured with a DuoSet IC Phosphatase Assay (R&D Systems) by determining the amount of free phosphate after incubation with a synthetic phosphopeptide substrate for 30 min, according to the manufacturer’s protocol. In some experiments, cells were pretreated for 10 min with increasing concentrations (10 to 100 µmol/L) of the antioxidant TEMPO before stimulation with 100 ng/mL SCF for 15 min.

### 4.8. Measurement of Intracellular ROS

Intracellular ROS levels were measured with the OxiSelect^™^ Intracellular ROS Assay kit (Cell Biolabs, Inc., San Diego, CA, USA). BMMC (2 × 10^5^ cells/1.5 mL) were incubated with 2′, 7′-dichlorofluorescin diacetate (DCFH-DA) (20 μM) for the last 20 min of stimulation with SCF at 37 °C and then washed with cold-PBS. Cells were lysed in 100 μL of 0.5% Triton-X 100 and the oxidation of DCFH into a fluorescent derivative (DCF) was measured in a fluorescence plate reader (Perkin Elmer, Shelton, CT, USA) at 492 nm excitation/535 nm emission and interpolated in a standard curve using H_2_O_2_ as the oxidative agent.

### 4.9. Statistical Analysis

Data were expressed as the mean ± SEM of values from at least three independent experiments performed in 3 to 5 separate BMMC cultures. A Student’s t-test was used to determine statistically significant differences between groups using Prism 8 software (Graph Pad Software, San Diego, CA, USA). Statistical significance was indicated as follows: * *p* < 0.05 and ** *p* < 0.01.

## Figures and Tables

**Figure 1 ijms-20-06216-f001:**
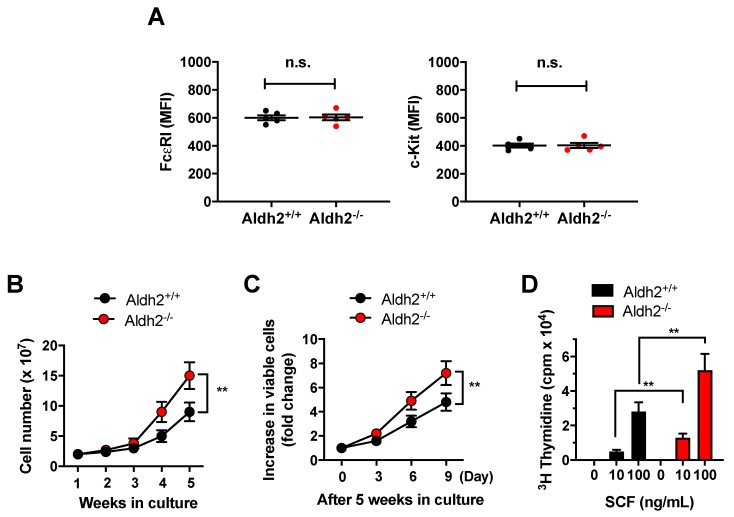
Aldehyde dehydrogenase 2 (Aldh2) deficiency promotes the proliferation of bone marrow-derived mast cells (BMMCs). (**A**) Mean fluorescence intensity (MFI) of cell surface FcεRI (left) and Kit (right) in BMMCs from *Aldh2^+/+^* and *Aldh2^−/−^* mice cultures grown for 5 weeks and analyzed concurrently. (**B**) Numbers of viable BMMCs from *Aldh2^+/+^* and *Aldh2^−/−^* mice at the indicated times in culture. Cells were stained with trypan blue and counted using a hemocytometer. (**C**) Increase in numbers of *Aldh2^+/+^* and *Aldh2^−/−^* mature mast cells (5 weeks old), plated at the same density, for 9 days in full media. (**D**) Proliferation of 5-week-old *Aldh2^+/+^* and *Aldh2^−/−^* mast cells measured by [^3^H]-thymidine incorporation. Cells were plated at the same density in media with or without the indicated concentrations of stem cell factor (SCF) for 24 h. Data are the mean ± SEM of five independent cultures. ** *p* < 0.01; n.s., not significant.

**Figure 2 ijms-20-06216-f002:**
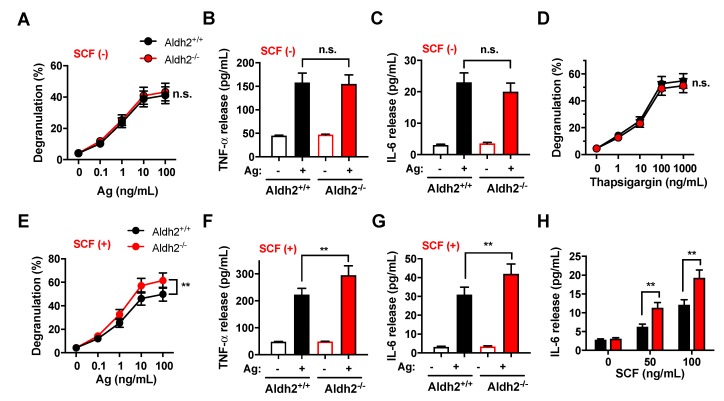
Aldh2 deficiency enhances mast cell responses to FcεRI and Kit co-stimulation, but not to FcεRI alone. (**A**–**C**) β-Hexosaminidase release (degranulation) (**A**), and release of the cytokines TNFα (**B**) and IL-6 (**C**), into the media in response to FcεRI stimulation. *Aldh2^+/+^* and *Aldh2^−/−^* BMMCs were sensitized with anti-Dinitrophenyl (DNP) IgE (100 ng/mL) overnight, washed, and then challenged with the indicated concentrations of Ag (DNP-HSA) in (**A**) or with 25 ng/mL of antigen (Ag) in (**B**,**C**). (**D**) Degranulation induced by the indicated concentrations of thapsigargin in *Aldh2^+/+^* and *Aldh2^−/−^* BMMCs. (**E**–**G**) Degranulation (**E**), and the release of TNFα (**F**) and IL-6 (**G**), induced by the co-stimulation of FcεRI and Kit in *Aldh2^+/+^* and *Aldh2^−/−^* BMMCs. Sensitized BMMCs were challenged with the indicated concentrations of Ag (**E**) or with 25 ng/mL of Ag in (**F**,**G**) in the presence of 100 ng/mL SCF. (**H**) IL-6 released by *Aldh2^+/+^* and *Aldh2^−/−^* BMMCs stimulated only with SCF at the indicated concentrations. Degranulation in (**A**,**D**,**E**) is expressed as the percentage of β-hexosaminidase released into the media compared to the total cellular content. Data are the mean ± SEM of five independent cultures. ** *p* < 0.01; n.s., not significant.

**Figure 3 ijms-20-06216-f003:**
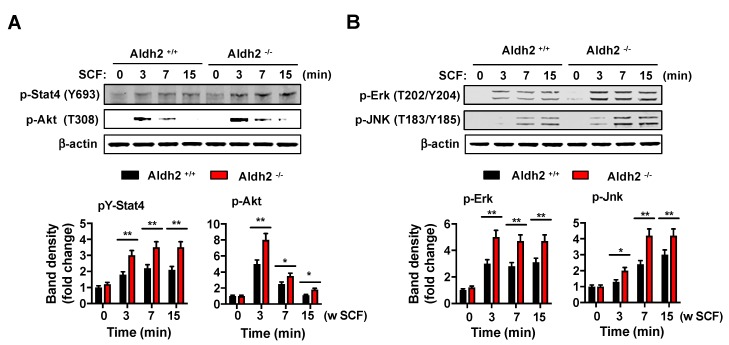
Aldh2-deficient BMMCs have enhanced Kit-mediated signaling. Changes in Stat4 and Akt phosphorylation (**A**) or p-Erk and p-Jnk (**B**) after stimulation with 100 ng/mL SCF for various times in *Aldh2^+/+^* and *Aldh2^−/−^* BMMCs, as indicated. The histogram below shows the average fold changes in band intensities after normalization, using β-actin as a loading control. Data are the mean ± SEM of three independent cultures. * *p* < 0.05; ** *p* < 0.01.

**Figure 4 ijms-20-06216-f004:**
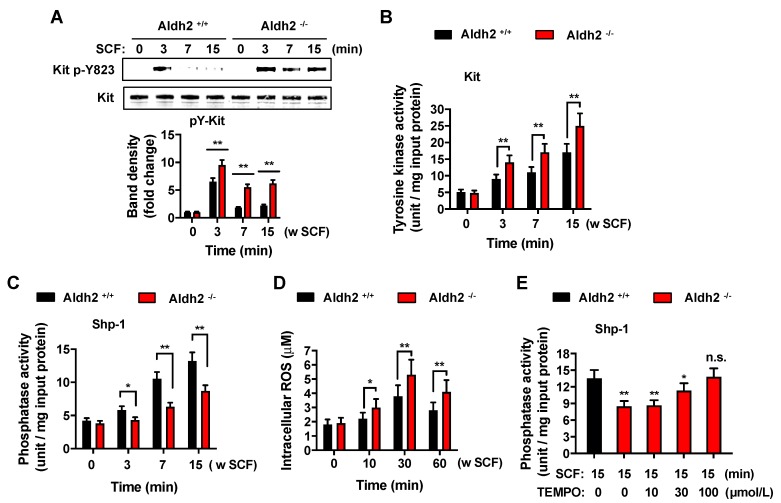
Aldh2 deficiency in BMMCs enhances the phosphorylation and activity of Kit and concomitantly increases ROS and reduces Shp-1 activity. Changes in Kit phosphorylation in cell lysates (**A**) and activity in Kit immunoprecipitates (**B**) after stimulation with 100 ng/mL SCF for various times in *Aldh2^+/+^* and *Aldh2^−/−^* BMMCs, as indicated. (**C**) Shp-1 activity in immunoprecipitates from *Aldh2^+/+^* and *Aldh2^−/−^* BMMCs stimulated with 100 ng/mL SCF for the indicated times. (**D**) Intracellular ROS levels in *Aldh2^+/+^* and *Aldh2^−/−^* BMMCs treated with 100 ng/mL SCF for the indicated times. Data in (**A**–**D**) are the mean ± SEM of three independent cultures. (**E**) Treatment with increasing concentrations of the antioxidant TEMPO reverses the reduced Shp-1 activation in SCF (100 ng/mL)-stimulated *Aldh2^−/−^* BMMCs. Data are the mean ± SEM of three independent experiments. * *p* < 0.05; ** *p* < 0.01; n.s; not significant.

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
