# Peer review of "Aldh2 Attenuates Stem Cell Factor/Kit-Dependent Signaling and Activation in Mast Cells"

_ijms, 2019, doi:10.3390/ijms20246216_

Round 1

Reviewer 1 Report

This is a well-designed study by Do-Kyum Kim et al showing that mitochondrial aldehyde dehydrogenase (ALDH2) deficiency in bone-marrow derived mast cells contributes to enhanced Kit tyrosine kinase phosphorylation and activity after stimulation with stem cell factor (SCF). The authors also provide evidence that ALDH2-deficiency is associated with increased ROS production and the reduced activity of the phosphatase Shp-1. Reduced Shp-1 activity seems to be concomitant with the sustained Kit signalling. 

Minor comments

1-I would only suggest to the authors to provide with more background in their Discussion on why the reduced activity of Shp-1 in ALDH2-deficient mast cells does not have an impact on IgE/Ag-mediated responses without SCF. 

2-Did the authors attempt to block/reduce ROS production in ALDH2-deficient mast cells to reverse their phenotype? This could significantly strength their hypothesis that increased ROS production is linked to reduced Shp-1 activity in ALDH2-deficient mast cells. 

Author Response

This is a well-designed study by Do-Kyum Kim et al showing that mitochondrial aldehyde dehydrogenase (ALDH2) deficiency in bone-marrow derived mast cells contributes to enhanced Kit tyrosine kinase phosphorylation and activity after stimulation with stem cell factor (SCF). The authors also provide evidence that ALDH2-deficiency is associated with increased ROS production and the reduced activity of the phosphatase Shp-1. Reduced Shp-1 activity seems to be concomitant with the sustained Kit signalling. 

We thank the reviewer for the positive comments.

Minor comments

1-I would only suggest to the authors to provide with more background in their Discussion on why the reduced activity of Shp-1 in ALDH2-deficient mast cells does not have an impact on IgE/Ag-mediated responses without SCF. 

We have now clarified this in the discussion (pg 6, lines 221-226). The data suggest that the effect of Aldh2 on Shp-1 activity occurs after SCF stimulation, but not after FceRI-ligation, since signaling events such as LAT, known to be targeted by Shp-1 in FceRI- signaling, and its downstream targets were not affected by Aldh2 deficiency (Figure S1).  The reasons why Shp-1 may not be regulated by Aldh2 after FceRI ligation are unclear at the moment.  Among possible explanations for the differences in Shp-1 regulation are that Aldh2 may contribute to the regulation of ROS induced by SCF but not by FceRI, or that only certain pools (or locations) of Shp-1 in complexes with other signaling proteins are affected by Aldh2.

2-Did the authors attempt to block/reduce ROS production in ALDH2-deficient mast cells to reverse their phenotype? This could significantly strength their hypothesis that increased ROS production is linked to reduced Shp-1 activity in ALDH2-deficient mast cells. 

We now provide a new figure (Figure 4E) showing that pre-incubation with a general antioxidant (TEMPO) reverses the reduced Shp-1 activity in Aldh2-/- cells in a dose-dependent manner, implicating reactive oxygen species in the diminished activation of Shp-1 by SCF in Aldh2-deficient cells. The description of the results and methodological details are included on pages 5, 6 and 8 (lines 176-177; 185-187; 218-219; and 319-320).

Reviewer 2 Report

The authors demonstrated that lack of Aldh2 in bone marrow-derived cultured mast cells should result in the attenuated responses induced by stem cell factor (SCF). The experiments were well designed, performed, and presented. Because accumulating evidence has indicated significant differences between cutaneous mast cells and cultured bone marrow-derived mast cells, it might be difficult to discuss alcohol-related flushing in detail based on their findings. Furthermore, several concerns described below should be addressed before publication.

Enhanced c-kit signaling in Aldh2-deficient mast cells should result in increased numbers of tissue mast cells and enhanced granule maturation in the Aldh2-/- mice, because SCF plays critical roles in proliferation and maturation of connective tissue-type mast cells. The authors should demonstrate or describe the characteristics of tissue mast cells of the Aldh2-/- mice, which will enhance the significance of this study.

Production of IL-6 was found to be rather decreased when BMMCs were stimulated with the combination of IgE/antigen and SCF. Figure 2C and 2H indictaed that the mounts of IL-6 production are ~25 ng/ml and ~50 ng/ml in the atigen- and SCF-stimulated cells, respectively, whereas Figure 2G indicated the amount is ~30 ng/ml in the cells stimulated with the combination. Accumulating evidence suggests that SCF should augment the antigen-induced IL-6 production. The authors should discuss this inconsistency.

Did tyrosine kinase activity of the immunoprecipitate with the anti-c-kit antibody reflect the activity of tyrosine phosphatses co-precipitated? Does the result in Figure 4B mean that the enzymatic activity of c-kit was increased in Aldh2-deficient BMMCs?

Line 80, "Thus, Aldh2 is not required for mouse mast cell maturation in vitro." The authors should moderate the tone because "maturation" means various aspects of cellular changes. They only demonstrated that the surface expression levels of two mast cell markers were unchanged. SCF is a well known regulator of mast cell maturation.

Line 117, Figure 1 A-C -> Figure 2 A-C

The values of the standard deviation are quite small in the experiments of cytokine measurements. The authors should present whether they repeated the experiments using BMMCs derived from different mice.

Author Response

The authors demonstrated that lack of Aldh2 in bone marrow-derived cultured mast cells should result in the attenuated responses induced by stem cell factor (SCF). The experiments were well designed, performed, and presented. Because accumulating evidence has indicated significant differences between cutaneous mast cells and cultured bone marrow-derived mast cells, it might be difficult to discuss alcohol-related flushing in detail based on their findings. Furthermore, several concerns described below should be addressed before publication.

Enhanced c-kit signaling in Aldh2-deficient mast cells should result in increased numbers of tissue mast cells and enhanced granule maturation in the Aldh2-/- mice, because SCF plays critical roles in proliferation and maturation of connective tissue-type mast cells. The authors should demonstrate or describe the characteristics of tissue mast cells of the Aldh2-/- mice, which will enhance the significance of this study.

We have toned down the sentence in the abstract related to alcohol flushing, that now reads: our findings reveal a functional role for mast cell intrinsic Aldh2 in the control of Kit activation and Kit mediated responses, which may lead to a better understanding of mast cell reactivity in conditions related to ALDH2 polymorphisms”. We feel this sentence or others in the manuscript suggesting a potential link to phenomena associated to alcohol flushing are mild enough not to overstate the reach of the findings, as we understand the study is in cultured mast cells.

We have not determined the number of mast cells in tissues as the study focused on the characterization of the role of Aldh2 on mast cell responses to IgE and SCF stimulation in vitro. As the reviewer points out, in vivo studies are of interest and deserve further investigation, but we feel may be beyond the scope of this paper. The number and characteristics of mast cells in connective tissue can be dictated by SCF but also many other microenvironment signals and the resulting effects on mast cell numbers or maturity may be variable from mouse to mouse and/or not robust enough to detect differences due to the complexity of signals involved. Nevertheless, if the reviewer considers that the data must be provided, we will examine the appearance and numbers of mast cells in tissues.

Production of IL-6 was found to be rather decreased when BMMCs were stimulated with the combination of IgE/antigen and SCF. Figure 2C and 2H indictaed that the mounts of IL-6 production are ~25 ng/ml and ~50 ng/ml in the atigen- and SCF-stimulated cells, respectively, whereas Figure 2G indicated the amount is ~30 ng/ml in the cells stimulated with the combination. Accumulating evidence suggests that SCF should augment the antigen-induced IL-6 production. The authors should discuss this inconsistency.

Thank you for noting this.  The units in the Y-axis of Figures 2 C and G were not comparable to those in Figure 2H.  The conditions were also not comparable: in Figure 2H the incubation volume was smaller and we used greater number of cells due to low production of IL-6 in the presence of SCF alone.  We have now converted the measurements in comparable units and are all expressed as pg/mL of IL-6 released.  With SCF alone, production was about 12 pg/mL (new Figure 2H), and when co-stimulated with antigen and SCF around 30 pg/mL (Figure 2G). The data also shows, as expected, that when SCF is added together with antigen, production of cytokines is increased. For TNF: from 150 ng/mL in the absence of SCF to 200 ng/mL in the presence of SCF (compare Figure 2 F and 2B); and for IL-6: from 22 pg/mL in the absence of SCF to 30 pg/mL in the presence of SCF (compare Figure 2G and 2C). 

Did tyrosine kinase activity of the immunoprecipitate with the anti-c-kit antibody reflect the activity of tyrosine phosphatses co-precipitated? Does the result in Figure 4B mean that the enzymatic activity of c-kit was increased in Aldh2-deficient BMMCs?

In this experiment Kit was immunoprecipitated at the indicated times after stimulation and its tyrosine kinase activity measured using a tyrosine kinase assay.  The increased Kit activity in Aldh2-/- is likely to reflect an increase in net Kit activity since Fig 4A also shows an increase in phosphorylation in tyrosine 823 in the kinase domain of the receptor, which indicates an active conformation. Co-immunoprecipitation of other proteins other than the target of immunoprecipitation is usually not very efficient, and under the conditions for this particular assay, Shp-1 was not detected in the Kit immunoprecipitates. Thus, we do not believe that the increased tyrosine kinase activity detected in the assay is a reflection of less phosphatase activity in the pull down, but a greater Kit activity.  However, the overall data is consistent with the conclusion that the higher Kit phosphorylation (detected by Western blotting) and activity (detected by the kinase assay) are a consequence of reduced Shp-1 activity in Aldh2-/- BMMC.   

Line 80, "Thus, Aldh2 is not required for mouse mast cell maturation in vitro." The authors should moderate the tone because "maturation" means various aspects of cellular changes. They only demonstrated that the surface expression levels of two mast cell markers were unchanged. SCF is a well known regulator of mast cell maturation.

We agree with the reviewer and have removed the sentence, which is not really needed.

Line 117, Figure 1 A-C -> Figure 2 A-C

Thank you.  This has been corrected

The values of the standard deviation are quite small in the experiments of cytokine measurements. The authors should present whether they repeated the experiments using BMMCs derived from different mice.

In these experiments, 5 cultures per genotype (i.e. from separate mice) were simultaneously grown.  The experiments for cytokines assays and degranulation were performed all at the same time and they were repeated at least three times.